Impact of extreme drought and incentive programs on flooded agriculture and wetlands in California’s Central Valley

Reiter Matthew E. 1 mreiter@pointblue.org
Elliott Nathan K. 1
Jongsomjit Dennis 1
Golet Gregory H. 2
Reynolds Mark D. 3
1 Point Blue Conservation Science , Petaluma, CA , USA
2 The Nature Conservancy , Chico, CA , USA
3 The Nature Conservancy , San Francisco, CA , USA
Esteban María Ángeles
Electronic publication date: 2018 Jun 29
Publication date: 2018
Volume: 6
Electronic Location ID: e5147
Received 2018 Feb 1; Accepted 2018 Jun 11
Copyright: © 2018 Reiter et al.
Copyright year: 2018
Copyright holder: Reiter et al.
License: This is an open access article distributed under the terms of the Creative Commons Attribution License, which permits unrestricted use, distribution, reproduction and adaptation in any medium and for any purpose provided that it is properly attributed. For attribution, the original author(s), title, publication source (PeerJ) and either DOI or URL of the article must be cited.
License URL: https://creativecommons.org/licenses/by/4.0/

Keywords: Agriculture, California, Drought, Water, Wetlands, Waterbirds, Habitat incentive program, Central Valley

Funding: The Nature Conservancy, the United States Geological Survey, the United States Fish and Wildlife Service, NASA NNX17AG81G Funding for this project was provided by The Nature Conservancy, the United States Geological Survey, the United States Fish and Wildlife Service, NASA (Grant Number NNX17AG81G), and the S.D. Bechtel, Jr. Foundation. The Nature Conservancy provided funding for the project but also two biologists (Golet and Reynolds) participated in the analysis and writing and are co-authors on the manuscript.

==============================
Background

Between 2013 and 2015, a large part of the western United States, including the Central Valley of California, sustained an extreme drought. The Central Valley is recognized as a region of hemispheric importance for waterbirds, which use flooded agriculture and wetlands as habitat. Thus, the impact of drought on the distribution of surface water needed to be assessed to understand the effects on waterbird habitat availability.

Methods

We used remote sensing data to quantify the impact of the recent extreme drought on the timing and extent of waterbird habitat during the non-breeding season (July–May) by examining open water in agriculture (rice, corn, and other crops) and managed wetlands across the Central Valley. We assessed the influence of habitat incentive programs, particularly The Nature Conservancy’s BirdReturns and The Natural Resources Conservation Service’s Waterbird Habitat Enhancement Program (WHEP), at offsetting habitat loss related to drought.

Results

Overall, we found statistically significant declines in open water in post-harvest agriculture (45–80% declines) and in managed wetlands (39–60% declines) during the 2013–2015 drought compared to non-drought years during the period of 2000–2011. Crops associated with the San Joaquin Basin, specifically corn, as well as wetlands in that part of the Central Valley exhibited larger reductions in open water than rice and wetlands in the Sacramento Valley. Semi-permanent wetlands on protected lands had significantly lower (39–49%) open water in the drought years than those on non-protected lands while seasonal wetlands on protected lands had higher amounts of open water. A large fraction of the daily open water in rice during certain times of the year, particularly in the fall for BirdReturns (61%) and the winter for WHEP (100%), may have been provided through incentive programs which underscores the contribution of these programs. However, further assessment is needed to know how much the incentive programs directly offset the impact of drought in post-harvest rice by influencing water management or simply supplemented funding for activities that might have been done regardless.

Discussion

Our landscape analysis documents the significant impacts of the recent extreme drought on freshwater wetland habitats in the Central Valley, the benefits of incentive programs, and the value of using satellite data to track surface water and waterbird habitats. More research is needed to understand subsequent impacts on the freshwater dependent species that rely on these systems and how incentive programs can most strategically support vulnerable species during future extreme drought.

Introduction

The Central Valley of California is a region of hemispheric importance for waterbirds (Gilmer et al., 1996; Shuford, Page & Kjelmyr, 1998; Central Valley Joint Venture (CVJV), 2006). With 90% of the historically occurring natural wetlands in the Central Valley gone (Frayer, Peters & Pywell, 1989), agricultural crops that are flooded post-harvest and hydrologically-managed wetlands are essential resources for migratory waterbirds (Elphick & Oring, 1998; Dybala et al., 2017; Shuford & Dybala, 2017). However, provisioning these crops and wetlands as waterbird habitat is dependent on a highly managed water system governed by dams, canals, water control structures, and water rights (Hanak & Lund, 2012). Meanwhile, the availability of water is dynamic both within and among years (Reiter et al., 2015; Reynolds et al., 2017). Future projections suggest that the inter-annual variability in the amount of waterbird habitat may increase with time due to the complex interactions of climate and human water management, even if long-term declines in average precipitation are not projected to be substantial (Matchett & Fleskes, 2017), making it critically important to understand how to manage wetlands and incentive-based habitat programs through extremes.

Between 2013 and 2015, the Central Valley of California and a large part of the western United States sustained an extreme drought (Griffin & Anchukaitis, 2014; Robeson, 2015). Because California’s water is so highly managed, anthropogenic factors play a large role in determining when and where drought impacts appear on the landscape (Hanak & Lund, 2012). Further, drought status, as measured by changes in precipitation, within the Central Valley may be less important to the availability of water in the Valley than the amount of snow pack in the surrounding mountain ranges which are the source of the Valley’s water (Carle, 2009). Previous analyses highlighted that while drought conditions across California’s Central Valley may be observed as a reduction in surface water in the southern Central Valley in the year of the drought, often multiple years of drought are required to see changes in the northern portion of the Central Valley (Reiter et al., 2015). The recent extreme and multi-year drought affecting California provides opportunity to gain additional insights into how more prolonged and extreme variations in the hydrology of the Sacramento and San Joaquin River watersheds may influence the distribution of waterbird habitat. This is especially important given that the incidence of such extremes is projected to increase in the future (Snyder, Sloan & Bell, 2004; Matchett & Fleskes, 2017).

In response to the drought, water restrictions (e.g., Term 91: Stored Water Bypass Requirements) were put into place in the Central Valley in the fall of 2014. The effects on the distribution of surface water caused by water restrictions, increasing water costs (Howitt et al., 2014), and lack of precipitation, needs to be assessed to understand impacts on waterbird habitat availability. Concurrent with this recent drought was the implementation of two incentive programs to help offset the cost of flooding agricultural fields to provide wetland habitat for migratory waterbirds (i.e., The Nature Conservancy’s BirdReturns program (Reynolds et al., 2017; Golet et al., 2018); Natural Resources Conservation Service’s Waterbird Habitat Enhancement Program (WHEP; Strum, Sesser & Iglecia, 2014)). The extent to which these incentive programs offset habitat losses due to the drought is not known. BirdReturns focused specifically on shorebirds, providing habitat <10 cm deep, in September and October and then again February to early April. WHEP incentivized flooding from November to February, though unlike BirdReturns, did not have a target water depth, but then staggered the timing of draining of those fields starting 1 February and lasting to 21 February to provide habitat into March.

Previous analysis of Central Valley water availability during drought quantified the extent of open surface water in the Central Valley between July and December for 2000–2011 (Reiter et al., 2015). To better characterize the magnitude and impacts of the recent extreme drought and to assess the relative contribution of flooded habitat as the result of incentive programs, analyses of more recent data compared to longer-term estimates (2000–2011; Reiter et al., 2015) of water extent were needed. Hence, our objectives with this study were to: Quantify the impact of the extreme drought between 2013 and 2015 on the timing and extent of available waterbird habitat (flooded agricultural fields and managed wetlands) during the non-breeding season (July–May) across the Central Valley.

Assess the influence of two incentive programs, BirdReturns and WHEP, at offsetting waterbird habitat loss resulting from drought.

Methods and Materials

Study area

We considered the entire Central Valley Joint Venture (CVJV) primary planning region (Dybala et al., 2017) to be the focal area for this study. The CVJV is a coalition consisting of 21 State and Federal agencies, private conservation organizations and one corporation that collaborates to achieve the common goal of providing for the habitat needs of migrating and resident birds in the Central Valley of California; a region of international importance for migratory waterfowl (Anseriformes) and shorebirds (Charadriiformes; Central Valley Joint Venture (CVJV), 2006). We further divided up the region into five basins according to Shuford & Dybala (2017) and used only the Sacramento Valley Basin and the San Joaquin Basin for some analyses (Fig. 1). The Central Valley falls completely within the Great Valley ecoregion (Hickman, 1993), and extends >400 km north to south and up to 100 km east to west; bounded by the Sierra Nevada, Cascade, and California Coastal Range mountains. The Central Valley climate is generally cooler and wetter in the north (Sacramento Valley Basin) than in the south (San Joaquin Basin and Tulare Basin). Water allocation and use in the Central Valley is highly managed and the southern portion of the Valley often relies on water being transferred for use from the north through contractual agreements (“water transfers”; Hanak & Lund, 2012). Consequently, there is generally less flooded agriculture in the southern Central Valley and higher year to year variability in flooding compared to in the north (Reiter et al., 2015). The majority of surface water in the Central Valley originates from snow pack in the adjoining Sierra Nevada and Cascade mountains (Carle, 2009).

Figure 1 Map of the Central Valley study area in California, USA.

The Central Valley Joint Venture boundaries and five basins of the Central Valley: Sacramento Valley Basin, Sacramento-San Joaquin River Delta [Yolo-Delta], San Joaquin Basin, Suisun Basin, and Tulare Basin. Data source: USGS and Central Valley Joint Venture (U.S. Fish and Wildlife Service).

Data and models

We derived data on the distribution of open water (<30% vegetated) across the Central Valley for 1 July–15 May of the following year, using satellite imagery and the supervised classification remote sensing techniques of Reiter et al. (2015). We used Landsat 5 Thematic Mapper for the period of 2000–2011 and Landsat 8 Operational Land Imager and Thermal Infrared Sensor for the period of 2013–2015. Because these sensors have different numbers of bands and slightly different wavelengths within each band, we developed separate boosted regression tree models for each satellite (Elith & Leathwick, 2009). We used data combined from ground and aerial surveys (n = 10,221 for our Landsat 5 model and n = 27,058 for our Landsat 8 model) to develop our models and to compare the relative bias associated with the predictions from each model. To prevent classification bias influencing our inference in this analysis, we bias-corrected the estimates of open water by the average difference between the true and estimated open water calculated using the ground-truth data for each sensor. We used separate correction factors for wetlands, rice, corn, and other crops.

We evaluated the timing and extent of open water from July to May for the Central Valley across several waterbird habitat cover types (seasonal and semi-permanent wetlands, rice [Oryza spp.], corn [Zea mays], and other suitable field crops and row crops [e.g. Triticum spp.; Gossypium spp.; Solanum lycopersicum]; see Dybala et al., 2017). To derive the amount of water by specific cover types (and to ensure that changes in water were not the result of changes in base acreages of potential habitat), we used two layers for wetlands and for agriculture representing the early 2000s (Stralberg et al., 2011; Homer et al., 2007) and then more recent habitat maps (2007–2014; Petrik, Fehringer & Weverko, 2014; Data S10). We considered cover types that were the same in both time periods as the baseline for assessing the proportion of each cover type that was open water. We overlaid each of the open water layers (2000–2015) on the habitat base layer to derive the proportion of each cover type that was open water.

Because the dynamics of water in the Central Valley are often non-linear, we followed Dybala et al. (2017) and used generalized additive mixed models (GAMMs; Wood, 2006; Wood & Scheipl, 2014) to assess the influence of time of year, drought, precipitation, region, and protected status (for wetlands only as most agriculture land is privately owned and not protected) on the proportion of open water in selected cover types between 1 July and 15 May of the following year. We evaluated GAMMs separately for each cover type. We fit a set of five models to agricultural crop data for 2000–2015 and six models to wetland cover type data (see covariate descriptions below). We filtered our data to only include satellite images with <50% cloud cover and then weighted observations in the model by the percent that was cloud free (50–100%). We included a random effect of water year to account for correlation among observations within the same year and an individual observation random effect to control for extra-binomial overdispersion in the data.

We characterized the impact of annual water conditions and drought by considering our full 2000–2015 data set to include three sets of water years (year types): non-drought years 2000–2011 (2000, 2003–2006; 2010–2011), drought years 2000–2011 (2001–2002, 2007–2009), and extreme drought (2013–2015). We considered a drought year to include any water year designated as “drought” or “critical” by the State of California Department of Water Resources. The State’s criteria for “drought” or “critical” are based on the projected runoff (million acre feet) on 1 May (see http://cdec.water.ca.gov/cgi-progs/iodir/WSIHIST for details on the level for each classification and data access). We also considered all years combined 2000–2011 as the recent long-term average with which to compare with the 2013–2015 drought.

Because rainfall likely influences the extent of open water on the landscape, we evaluated models that included estimates of total precipitation. Rainfall data were taken from daily historic rainfall amounts recorded at weather stations via the NOAA National Climatic Data Center (www.ncdc.noaa.gov/cdo-web). To characterize rainfall across the Central Valley, we used data from one weather station in the northern and southern parts of the valley (Sacramento Metropolitan Airport and Fresno Yosemite International Airport) which had consistent temporal coverage across our study period. For each station, we calculated two- and four-week running totals then averaged these across stations. Precipitation data was then matched to the average date of the three main Landsat scenes covering the Central Valley for a given two-week period. Including precipitation in models allowed us to characterize the effect of recent localized rainfall in creating habitat, rather than broader scale water allocation decisions, and specifically if there were differences in the effect of rainfall across cover types. We hypothesized that agriculture cover types would be more likely to show a precipitation signature as there are many hectares that are not flooded July–May, whereas a larger fraction of managed wetlands are already flooded by mid-to-late winter (Dybala et al., 2017) when precipitation is expected to have its greatest impact.

To assess the impact of drought on private versus protected area wetlands, where differences might influence conservation and management strategies, we considered protection status as a single-factor in models and allowed an interaction with the year type. We derived protection status using the California Protected Areas Database (CPAD) (GreenInfo Network, 2016) overlaid on the habitat cover type layer to identify wetland cover types that fell within a protected area. The California Protected Area Database defines protected areas as those that are owned in fee and managed for open space purposes. Any cover types that fell outside of a protected area were assumed to be private or not protected. Since nearly all agriculture is on private land, we did not evaluate the influence of protection for models of rice, corn, or other crops.

To be able to better understand how within-year temporal availability of open water might differ among years in this dynamic system and given that interactions with smoothing terms (herein, day) are hard to include in GAMMs, we also fit separate GAMMs for each of the three year types (non-drought 2000–2011, drought 2000–2011, extreme drought 2013–2015) in each cover type with only a smoothing parameter of day. We plotted the model fitted values and 95% CI for each of the three data sets for each crop type to evaluate variation in the magnitude of the differences through the year. Covariates for precipitation and land protection were not included in these year type specific models.

To characterize spatially variability in the impact of drought on wetlands, we compared the amount and timing of open water in seasonal wetlands between the Sacramento Valley Basin and San Joaquin Basin. We compared these two basins of the Central Valley because they have the most extensive managed wetlands and previous analyses have shown differences in the impact of drought between the two regions (Reiter et al., 2015). We fit separate GAMMs to seasonal wetland data from each basin that compared all three year-type groups (non-drought 2000–2011, drought 2000–2011, extreme drought 2013–2015).

Incentive programs

To quantify the relative contribution of the BirdReturns and WHEP habitat incentive programs, we calculated the proportion of the total estimated flooded rice habitat in the Sacramento Valley Basin that was provided by these programs. Specifically, we evaluated the contribution of fields that were flooded between July 2013 and May 2014 and again between July 2014 and May 2015. We compared the relative contribution of these programs to the habitat available in rice during the extreme drought (2013–2015) as a measure of the relative return on investment.

As the incentive programs largely focused on the Sacramento Valley basin and only in rice agriculture, we developed a GAMM of rice flooding using a subset of the data for that geography for the comparison (Fig. 1). We considered a combined model for the 2013–2015 data that included a smoothing term of day of the year relative to 1 July and an observation level random effect to account for overdispersion. We multiplied the daily model-fitted estimate of the proportion flooded by the estimated amount (ha) of rice planted in each year (216,105 ha in 2013 and 169,606 ha in 2014; Dybala et al., 2017) to get the area that was open water in each day. We then calculated the proportion of total habitat available per day provided by BirdReturns and WHEP in each of the year sets. Because the duration of flooding can influence the value of the habitat, we considered a metric “habitat ha days” for additional comparisons. Habitat ha days was the sum of all flooded ha across all days in the year. Each flooded ha on a day contributes one habitat ha day to the calculation.

To account for habitat remaining in rice fields upon termination of incentivized flooding, we used data from another study in rice to estimate the average duration that water remained in fields during the period that fields were drained (i.e., drawdown). These data are observations from fields visited two times per week until dry following the initiation of draining (Data S8). Water depth, the percent flooded, and the percent saturated was estimated for each field. We fit a GAMM to estimate the probability that an individual field would have waterbird habitat as a function of the days since the initiation of draining the field using observations from February to March in 2012 and 2013. We defined habitat as present in the observation data if fields had water depth >0 cm, or if fields were >0% flooded, or if fields were >50% saturated. Because field data included repeated visits to individual fields, we considered field as a random effect in the model. We multiplied the model fitted probability of habitat by day since draining was initiated with the amount of habitat when the draining started to estimate the daily amount of habitat remaining. We evaluated both a minimum estimate of habitat provided (assumes no habitat provided during drawdown following of end of practice) as well as the model corrected estimates.

Model fit and effect size

Overall, we evaluated relative model fit for each cover type and analysis using adjusted-R2 and considered coefficients with P < 0.05 to be significant and P < 0.10 to be marginally significant. We characterized the effect size of covariates in our logistic GAMMs using the odds for individual effects (Zuur et al., 2009). Specifically, we calculated the percent change by of drought years over non-drought or average years 2000–2011 in our models as (eBxi − 1) * 100; where Bxi is the coefficient estimate for factor x, level i. All statistical analyses were completed using R v.3.3 (R Core Team, 2017) and the “gamm4” package (Wood & Scheipl, 2014).

Results

Assessment of water classification models suggested some subtle differences in bias between our Landsat 5 and Landsat 8 derived data (Table 1). Overall, across cover types the Landsat 8 model was more accurate. Among cover types, open water in freshwater emergent wetlands was predicted with the lowest accuracy by both satellites. Only in the case of corn did the directionality of the bias differ between the sensors. We used these cover type specific values to correct our observed estimates from each classification model.

Table 1 Landsat 5 ETM and Landsat 8 OLI water classification model accuracy and bias for the Central Valley of California.

Satellite	Cover type	N	Accuracy	Bias	
Landsat 8	Corn	2,237	0.95	0.05	
Landsat 5	Corn	46	0.89	−0.07	
Landsat 8	Rice	2,756	0.94	0.04	
Landsat 5	Rice	640	0.89	0.03	
Landsat 8	Other	1,005	0.99	0.001	
Landsat 5	Other	475	0.96	0.003	
Landsat 8	Freshwater emergent wetland	1,765	0.88	−0.11	
Landsat 5	Freshwater emergent wetland	5,564	0.79	−0.01	
Notes:

Accuracy (proportion correctly classified) and bias (average difference between the true and estimated probability of open water) of estimates of open water by different satellites (Landsat 5 ETM and Landsat 8 OLI) in three crop types (rice, corn, other crops [field crops, row crops, grain crops]) and managed wetlands (seasonal and semi-permanent).

Models for open water in rice, corn, and seasonal and semi-permanent wetlands were a good fit to the data based on residuals and explained 30–79% of the variance (Table 2). Models for other crops consistently explained relatively less of the variation. There were significant declines in open surface water during the recent extreme drought (2013–2015) in all cover types evaluated except for semi-permanent wetlands (Tables 3 and 4). In the agricultural landscape, the recent drought resulted in significantly less open water than the non-drought or average years; 45–57% declines in rice, 77–81% declines in corn, and 64–71% declines in other crops. However, after accounting for precipitation (two-weeks or four-weeks), which had a significant or marginally significant positive effect on open water in all crops (Fig. 2; Table 3), the difference between drought and non-drought years was typically less though still statistically significant (Table 3).

Table 2 Adjusted-R2 values for models of the proportion open water in three crop types and two managed wetland types in the Central Valley of California 2000–2015.

Model	Rice	Corn	Other	Seasonal	Semi-permanent	
Day1 + Year type2	0.62	0.28	0.15	0.79	0.56	
Day + Year type * Protection3 + Precip2wk4	0.63	0.29	0.15	0.79	0.59	
Day + Year type * Protection + Precip4wk5	0.62	0.30	0.36	0.79	0.59	
Day + Extreme drought6	0.63	0.29	0.14	0.79	0.56	
Day + Extreme drought * Protection + Precip2wk	0.64	0.28	0.15	0.79	0.59	
Day + Extreme drought * Protection + Precip4wk	0.62	0.30	0.36	0.79	0.59	
Notes:

Generalized additive mixed models were fit to assess the proportion of open water in three crop types (rice, corn, other crops [field crops, row crops, grain crops]) and two managed wetland types (seasonal and semi-permanent). Adjusted-R2 values indicate what proportion of the variance in the data was explained by the model. The protection variable was not included in crop type models.

1 Day = indicator for day of the year between 1 and 319 starting as July 1 = 1.

2 Year type = non-drought 2000–2011; drought 2000–2011; extreme drought 2013–2015.

3 Protection = factor indicating whether the land is under protected status; not considered models of rice, corn or other crops.

4 Precip2wk = total precipitation measured for two-weeks before the open water estimate from Landsat.

5 Precip4wk = total precipitation measured for four-weeks before the open water estimate from Landsat.

6 Extreme drought = factor indicating data from years 2013 to 2015.

Table 3 Coefficient estimates (β) and model estimated percent annual change (%) in the proportion of open water in rice, corn, and other crops in the Central Valley of California 2000–2015.

		Rice		Corn		Other	
Model	Covariate	β	SE	%		β	SE	%		β	SE	%	
1	Non-drought	−2.38	0.15			−2.97	0.11			−3.58	0.08		
	Drought	−0.14	0.23	−13		−0.21	0.17	−18		−0.14	0.13	−13	
	Extreme drought	−0.85	0.27	−57		−1.63	0.21	−80		−1.25	0.16	−71	
2	Non-drought	−2.57	0.16			−2.90	0.12			−3.62	0.09		
	Drought	−0.12	0.21	−11		−0.21	0.17	−19		−0.15	0.13	−13	
	Extreme drought	−0.67	0.29	−49		−1.71	0.22	−82		−1.22	0.17	−71	
	Precip two-weeks	1.83	0.41			−1.08	0.65			0.51	0.31		
3	Non-drought	−2.65	0.16			−3.04	0.13			−3.79	0.09		
	Drought	−0.09	0.22	−8		−0.20	0.17	−18		−0.11	0.14	−10	
	Extreme drought	−0.65	0.28	−48		−1.56	0.22	−79		−1.06	0.17	−65	
	Precip four-weeks	1.10	0.32			0.34	0.37			1.12	0.17		
4	Average	−2.44	0.12			−3.06	0.09			−3.64	0.07		
	Extreme drought	−0.79	0.32	−54		−1.54	0.20	−79		−1.17	0.17	−70	
5	Average	−2.62	0.12			−2.99	0.10			−3.68	0.07		
	Extreme drought	−0.62	0.27	−46		−1.62	0.21	−80		−1.17	0.17	−69	
	Precip two-weeks	1.85	0.34			−1.07	0.65			0.51	0.31		
6	Average	−2.69	0.14			−3.13	0.11			−3.84	0.07		
	Extreme drought	−0.61	0.30	−45		−1.48	0.21	−77		−1.02	0.17	−64	
	Precip four-weeks	1.01	0.35			0.36	0.37			1.13	0.17		
Notes:

Coefficient estimates from generalized additive mixed models for drought 2000–2011 and extreme drought 2013–2015 should be interpreted relative to the intercept term of non-drought 2000–2011 (Models 1–3) and average 2000–2011 (Models 4–6). Estimates in bold were statistically significant with P < 0.05 and those in italics were considered marginally significant with P < 0.10.

Table 4 Coefficient estimates (β) and model estimated percent annual change (%) in the proportion of open water in seasonal and semi-permanent wetlands in the Central Valley of California 2000–2015.

		Seasonal		Semi-permanent	
Model	Covariate1	β	SE	%		β	SE	%	
1	Non-drought	−1.02	0.10			−0.07	0.15		
	Drought	−0.01	0.16	−1		0.06	0.24	7	
	Extreme drought	−0.68	0.20	−49		−0.34	0.30	−29	
2	Non-drought	−0.93	0.12			0.24	0.17		
	Drought	−0.09	0.19	−9		0.15	0.26	17	
	Extreme drought	−0.93	0.23	−60		−0.41	0.33	−34	
	Protected	−0.13	0.15	−12		−0.49	0.17	−39	
	Protected * Drought	0.16	0.20	17		−0.14	0.26	−13	
	Protected *  Extreme drought	0.52	0.26	68		0.03	0.34	2	
	Precip two-weeks	−2.83	2.13			−8.74	4.47		
3	Non-drought	−0.97	0.14			0.16	0.18		
	Drought	−0.09	0.19	−8		0.16	0.27	17	
	Extreme drought	−0.90	0.23	−59		−0.38	0.34	−32	
	Protected	−0.13	0.13	−13		−0.50	0.17	−40	
	Protected * Drought	0.16	0.20	18		−0.14	0.26	−13	
	Protected * Extreme drought	0.51	0.26	67		0.04	0.35	4	
	Precip four-weeks	0.83	1.65			0.28	2.36		
4	Average	−1.02	0.08			−0.04	0.12		
	Extreme drought	−0.68	0.19	−49		−0.36	0.30	−31	
5	Average	−0.96	0.09			0.30	0.13		
	Extreme drought	−0.89	0.22	−59		−0.48	0.31	−38	
	Protected	0.45	0.24	57		0.08	0.33	9	
	Protected * Extreme drought	−0.07	0.10	−7		−0.55	0.13	−43	
	Precip two-weeks	−2.81	1.67			−8.73	4.14		
6	Average	−1.01	0.10			0.22	0.14		
	Extreme drought	−0.87	0.22	−58		−0.44	0.32	−36	
	Protected	0.45	0.25	56		0.09	0.33	10	
	Protected * Extreme drought	−0.07	0.10	−7		−0.56	0.13	−43	
	Precip four-weeks	0.81	1.55			0.27	2.31		
Notes:

Coefficient estimates from generalized additive mixed models for drought 2000–2011 and extreme drought 2013–2015 should be interpreted relative to the intercept term of non-drought 2000–2011 (Models 1–3) and average 2000–2011 (Models 4–6). Estimates in bold are statistically significant with P < 0.05 and those in italics P < 0.10.

1 Protected * Drought and Protected * Extreme drought represent the influence of protected lands in drought and extreme drought years.

Figure 2 Estimated proportion open water in rice on 15 January in the Central Valley of California when accounting for precipitation.

“No Precip” (black) assumes no rain falls in the previous four-weeks whereas “Precip” (blue) assumes average rainfall. Open water estimates were derived from generalized additive mixed models fit to all water distribution data from 2000 to 2015. Variables in the model included the amount of precipitation within four-weeks of the observed date, an indicator for drought years (2000–2011 and 2013–2015 (extreme drought), and a smoothing parameter for day of the year (July 1 = 1)). Fitted means are plotted with 95% confidence intervals.

While seasonally flooded managed wetlands showed significant declines in open water in the recent drought compared to historic non-drought (47–58%) and average years (49–59%; Table 4), changes in semi-permanent wetlands were not significant, though all estimates for drought variable coefficients were negative. Precipitation did not have a significant effect on managed wetlands, however, between 2000 and 2015 seasonal and semi-permanent wetlands had different amounts of open water with respect to protected areas during drought years. Semi-permanent wetlands had significantly less open water on protected land (∼40%) compared to non-protected areas whereas seasonal wetlands had marginally significant (P < 0.10) more open water on protected land (∼56%) than on non-protected land (Table 4). The effects of protected land in wetlands appeared to be magnified during the recent drought with significant interactions detected between protection and extreme drought years compared to non-drought years 2000–2011 and all years 2000–2011.

Modeling year types separately emphasized the temporal differences in the timing and amount of water among years though was not used for statistical inference. In particular, it highlighted the period with the largest reductions in open water generally occurring across all cover types October through March (Figs. 3 and 4). Other crops were particularly reduced November to March while in corn there was substantially reduced water in nearly all months (Fig. 3). In rice, the recent drought reduced open water in February and March but also in April and May. In seasonal and semi-permanent wetlands, the reduction in water was largely observed between October and March (Fig. 4).

Figure 3 Estimated proportion of (A) rice, (B) corn, and (C) other crops that was open water in the Central Valley of California between 1 July and 15 May based on data from 2000–2011 and 2013–2015.

Estimates were derived from separate models from separate generalized additive mixed models for each year group of non-drought 2000–2011, drought 2000–2011, and extreme drought 2013–2015. Models for each year group included only a smoothing parameter for day of year. Fitted means are plotted with 95% confidence interval bands.

Figure 4 Estimated proportion of (A) seasonal and (B) semi-permanent wetlands that was open water in the Central Valley of California between 1 July and 15 May based on data from 2000–2011 to 2013–2015.

Estimates were derived from separate generalized additive mixed models for each year group of non-drought 2000–2011, drought 2000–2011, and extreme drought 2013–2015. Models for each year group included only a smoothing parameter for day of year (July 1 = 1). Fitted means are plotted with 95% confidence interval bands.

Patterns of seasonal wetland inundation differed between the Sacramento Valley Basin and the San Joaquin Basin, as did the impact of drought (Fig. 5). Seasonal wetlands in the Sacramento Valley overall have a higher proportion of open water and experienced, on average, 63–69% declines in open water during the 2013–2015 extreme drought while the San Joaquin Basin had declines of 85–86% (Table 5). Additionally, the San Joaquin Basin showed evidence of a lower but more prolonged peak in open water than the Sacramento Valley in both drought and non-drought years (Fig. 5).

Figure 5 Estimated proportion of seasonal wetlands that was open water in the (A) Sacramento Valley Basin and (B) San Joaquin Basin of California 2000–2015.

Estimates were derived separately for these two regions which have the largest amount of managed wetlands using a single model for each region and a factor for each year type: non-drought 2000–2011, drought 2000–2011, and extreme drought 2013–2015. A smoothing parameter for day of year (July 1 = 1) was also included in models. Fitted means are plotted with 95% confidence interval bands. Dark green areas represent overlap between confidence bands.

Table 5 Summary of the proportion of open water in seasonal wetlands in the Sacramento Valley and the San Joaquin Valley in the Central Valley of California 2000–2015.

Region	R2	Covariate	Estimate	SE	P	%	
Sacramento	0.63	Non-drought	−1.41	0.20			
		Drought	−0.01	0.31	0.98	−1	
		Extreme drought	−1.02	0.43	0.02	−64	
San Joaquin	0.46	Non-drought	−1.48	0.37			
		Drought	−0.49	0.58	0.40	−39	
		Extreme drought	−2.16	0.70	0.01	−88	
Notes:

Adjusted-R2, coefficient estimates (β), and estimated percent change in open water (%) from generalized additive mixed models are presented. Coefficient estimates from for drought 2000–2011 and extreme drought 2013–2015 should be interpreted relative to the intercept term of non-drought 2000–2011. Estimates in bold were statistically significant with P < 0.05.

The effect of incentive programs was noticeable when looking at flooding in rice in the Sacramento Valley (Fig. 6). The total area incentivized as part of BirdReturns in the region was 4,980 ha in spring 2014, 2,759 ha for fall 2014, and 1,357 ha for spring 2015 (Golet et al., 2018). Given the timing and duration of practices, this resulted in a minimum estimated total of 168,022 habitat ha days between 1 February and 4 April 2014 and, 85,666 habitat ha days between 1 September 2014 and 31 March 2015. WHEP incentivized 32,473 ha and 27,600 ha of habitat creation, respectively, in 2013–2014 and 2014–2015, which resulted in a minimum of 3.3 million habitat ha days (2013–2014) and 2.9 million habitat ha days (2014–2015) across the entire time period.

Figure 6 Estimated average daily percentage of open water in post-harvest rice provided by habitat incentive programs when active during the 2013–2015 extreme drought in the Central Valley of California.

Incentive programs were The Nature Conservancy’s BirdReturns and the Natural Resources Conservation Service’s Waterbird Habitat Enhancement Program (WHEP). Three seasons evaluated for 2013–2015 were 1 September to 31 October (Fall), 1 November to 31 January (Winter), and 1 February to 4 April (Spring). Note: WHEP was only active 1 February to 7 March in late-winter and spring as fields were drained.

Our model to characterize presence of habitat during drawdown suggested that there was a significant negative effect of days since draining on the probability of habitat presence. However, there was a greater than zero probability of waterbird habitat for up to 30 days after the end of the incentivized practice and the initiation of draining. This indicated the end of the incentivized period does not immediately end the habitat value. After accounting for a slow drawdown of water once the practices were complete by including our model-based estimates of proportion of remaining habitat for 30 days post-drawdown, BirdReturns provided an estimated total of 221,072 habitat ha days occurring between 1 February and 4 May 2014 (adds 30 days to latest end date of practice) and 128,046 habitat ha days between 1 September 2014 and 30 April 2015, while WHEP provided an estimated 3.7 million habitat ha days in 2013–2014 and 3.1 million habitat ha days in 2014–2015. On days when the program was active between 1 September and 31 October 2014, BirdsReturns provided 14–61% (mean = 39%) of the daily waterbird habitat in flooded rice fields (Fig. 6). In the spring (2014: 1 February–4 May; 2015: 1 February–28 April), BirdReturns provided proportionally less habitat than in fall with on average 5% per day (min = 1%, max = 13%). When active, WHEP, on average, provided 64% (Min = 33%, Max = 100%) of the daily flooded rice in the winter (1 November to 31 January) and 29% (min = 15%, max = 46%) between 1 February and 7 March.

Discussion

The extreme drought recently experienced in California impacted human, agricultural, and natural systems. Our study highlights that the drought caused a substantial reduction in open water habitats across the agricultural and wetland landscapes of the Central Valley and that the impact varied spatially and temporally. The observed decline ranged from approximately 20–86% depending on cover type, time of year, and region. Overall, post-harvest flooded rice declined less during the drought than flooded corn, other waterbird compatible crops, or seasonal wetlands. Further, seasonal wetlands in the San Joaquin Basin declined more than in the Sacramento Valley, confirming previous observations of spatial differences in the impact of drought across the Central Valley (Reiter et al., 2015).

The 2013–2015 drought reduced waterbird habitat over non-drought years more than previous droughts between 2000 and 2011, highlighting its severity. Estimates of open water for the 2013–2015 drought were lower than drought years between 2000 and 2011 across nearly all models and cover types. The length and severity of the recent extreme drought likely contributed to the observed decline as water restrictions were enacted and the cost of water began to increase (Howitt et al., 2014). More intensive modeling, however, is needed to tease out these policy and socio-economic drivers of changes in water applied to the landscape.

Mid-winter or peak flooding (November to February) appeared most affected across cover types. Fall, which is generally the driest time of the year (Reiter et al., 2015) and already a period of habitat limitation for migratory shorebirds (Dybala et al., 2017), remained dry across cover types evaluated, but did not show particularly significant reductions in open water during the drought. The flooding pattern was similar in spring, however open water in post-harvest rice declined very quickly and was particularly low March through May during the 2013–2015 drought compared to other drought and non-drought years. Open water in rice during April and May, which is associated with the planting of the rice crop, was also delayed during the drought, supporting previous findings of drought impacts on open water and exacerbating the mismatch in the timing of habitat for migratory birds (Schaffer-Smith et al., 2017). Overall open water in post-harvest rice experienced smaller declines compared with other crop types and seasonal wetlands. While this is consistent with previous work that highlights the resilience of the Sacramento Valley surface water compared to other regions (Reiter et al., 2015), our results also suggest that a large fraction of the open water in rice (up to 100% of observed) during certain times of the year, particularly fall and winter, may have been provided through incentive programs.

The value of incentive programs to generate habitat and ecosystem services in the Central Valley has been documented (Duffy & Kahara, 2011; DiGaudio et al., 2015; Golet et al., 2018), yet this is the first regional-scale assessment of the effectiveness and additionality of incentive programs in providing wetland habitat during drought and further underscores the contribution of these programs. BirdReturns was particularly effective at providing habitat in fall; a period that is already thought to be food limiting for migratory shorebirds (Dybala et al., 2017). Fields enrolled in BirdReturns during in fall 2014 had some of the highest shorebird densities ever reported for agriculture in this region, confirming this to be a time of habitat deficit (Golet et al., 2018). The WHEP was effective during the period of peak flooding when nearly 70% of available flooded rice habitat was provided by the program. However, it is not known what proportion of those individual farmers who enrolled in either BirdReturns or WHEP would have adopted the enhancement practices even if the incentive payments were not available (Baumgart-Getz, Prokopy & Floress, 2012; Reimer et al., 2014). Further assessment is needed to know how much the incentive programs directly offset the impact of drought in post-harvest rice or simply supplemented funding for activities that might have been done regardless.

Our analysis highlights that recent localized precipitation can help supplement the open water habitat in agriculture that is largely created through intentional diversions of snow melt from the surrounding mountain ranges (Central Valley Joint Venture (CVJV), 2006; Hanak & Lund, 2012; Golet et al., 2018). The strong positive effect of precipitation was most noticeable in agricultural cover types and particularly in rice. Some of the reduced flooding in rice in the recent drought compared to non-drought years may be the result of less rain or potentially less saturated soils from intentional flooding that can become open water with additional rainfall. While much of this precipitation-driven water detected using satellites, we assume, may be too shallow for most waterfowl (particularly ducks), it certainly has value for shorebirds, wading birds, and other freshwater dependent taxa (Strum et al., 2013). Further research is needed to determine the overall contribution to habitat of rain flooded agricultural fields and consequently how incentive programs may vary in effectiveness in wet versus dry years. However, our results also highlight that among year variation in habitat availability due to year type (drought versus non-drought) likely plays a bigger role in the amount and timing of habitat than within Central Valley precipitation.

While habitat availability appeared to decline substantially during some points of the year in certain cover types, our analysis does not directly assess the potential impacts to the wildlife that rely on these systems. Recent work by Petrie et al. (2016) indicated that the drought in the Central Valley could have had significant impacts on waterfowl populations. They used expert opinion to develop drought scenarios and a bioenergetics model to determine impact to waterfowl from a food energy perspective. The scenario they developed assumed a 25% decline in flooded wetlands in 2014–2015. However, our satellite and model derived estimates for the same period suggest a much more severe impact of the drought on wetlands than was assumed by Petrie et al. (2016). Parameterizing their bioenergetics model with data from this study could help to further illuminate the species and population level impacts of the drought. Similarly, a recently developed bioenergetics model for shorebirds could further assess the impacts of drought on these species which rely on open water cover types in wetlands and flooded agriculture (Dybala et al., 2017). However, integrating our data with bioenergetics models for waterfowl or shorebirds will require the development of two additional parameters for drought not evaluated here: changes in wetland moist soil seed productivity for waterfowl (Naylor, 2002) and changes in water depth profiles for shorebirds (Dybala et al., 2017).

Open water in seasonal wetlands declined significantly during the recent drought in both the Sacramento Valley and the San Joaquin Basin. However, the peak proportion of open water was higher in seasonal wetlands in the Sacramento Valley Basin and declined less during drought compared with the San Joaquin Basin. This spatial difference may be in part explained by the fact that seasonal wetlands on protected land had a higher proportion of open water than non-protected, largely private, seasonal wetlands during the recent extreme drought; a larger proportion of the managed wetlands in the San Joaquin Basin are privately owned compared to the Sacramento Valley. Beyond speculating on the observed patterns, we were unable to evaluate why there may have been differences in the impact of drought in seasonal wetlands based on protection status. Further complicating interpretation was the finding that semi-permanent wetlands had an opposite pattern with lower open water in protected wetlands.

We also quantified differences in the timing of open water in seasonal wetlands between the Sacramento Valley Basin and the San Joaquin Basin with the peak proportion of open water occurring earlier and remaining on the landscape longer in the San Joaquin Basin (November to March) compared to the Sacramento Valley (end of November to early March). While we do not know the exact cause of these different patterns, recent studies of overwintering shorebirds in the Central Valley suggest that shorebirds in the more hydrologically dynamic Sacramento Valley move longer distances and migrate out of the area significantly more than birds in the San Joaquin Basin (Barbaree et al., 2018). Differential patterns of wetland inundation may be driving some of these observed differences in movement ecology. Incorporating different flooding patterns among regions of the Central Valley into bioenergetics models (Petrie et al., 2016; Dybala et al., 2017) could inform strategies of how to maximize the value of the habitat created across the whole landscape for waterfowl and shorebirds.

Conclusion

Our study highlights the negative impacts that extreme drought can have on essential wetland and agricultural waterbird habitats in the Central Valley of California but also the substantial benefits that can be provided through habitat incentive programs. Climate change models and habitat projection scenarios for California indicate the strong likelihood of increasing temperatures and more, potentially extreme, variation in precipitation patterns (Snyder, Sloan & Bell, 2004; Matchett & Fleskes, 2017). With more limited water resources, our results suggest that wetland managers will need to be ever more strategic in how they allocate incentive program water to prevent the reductions observed in the recent extreme drought. Furthermore our assessment provides a novel perspective of the impacts of extreme drought in the Central Valley and points to the need to have dynamic strategies (Reynolds et al., 2017) to provide more resilient habitat in flooded agriculture and wetlands during early to late winter, in the face of additional, and potentially more extreme, drought events. Lastly, we conclude that remotely sensed data can be a powerful tool to track water in the Central Valley and should be harnessed to regularly update water and wetland managers on how much habitat is available and where, so that there can be more coordinated data-driven water management. While many sophisticated models of water scenarios can be evaluated (Draper et al., 2003; Yates et al., 2009), understanding where water and wetland habitats are ultimately distributed on the landscape in space and time is needed for water managers to make decisions that maximize the value of limited water resources for wildlife (DWR, 2009).

Supplemental Information

Supplemental Information 1 Look-up table for CVJV basins.

Click here for additional data file.

Supplemental Information 2 Data for analysis of open water in agriculture fields in the Central Valley.

Click here for additional data file.

Supplemental Information 3 Data used for bias correction of estimates of water from Landsat 5 and Landsat 8.

Click here for additional data file.

Supplemental Information 4 Data used for Sacramento Valley level assessment of flooding in rice as part of analysis of impact of incentive programs.

Click here for additional data file.

Supplemental Information 5 Data used to calculate the amount of habitat generated by incentive programs in the Central Valley.

Click here for additional data file.

Supplemental Information 6 Data used to assess open water in managed wetlands in the Central Valley.

Click here for additional data file.

Supplemental Information 7 R code for evaluating impact of incentive programs in the Central Valley.

Click here for additional data file.

Supplemental Information 8 Water depth data used to model residual habitat post initiation of field draining.

Click here for additional data file.

Supplemental Information 9 Code for calculating decay rate of water in field once draining of water is initiated.

Click here for additional data file.

Supplemental Information 10 The Nature Conservancy’s unpublished cover type data layer for the Central Valley.

Click here for additional data file.

Supplemental Information 11 R-code for analyzing impacts of drought on open water in managed wetlands and agriculture using single model for all year types.

Click here for additional data file.

Supplemental Information 12 R-code for analyzing wetland flooding in the Sacramento Valley and San Joaquin Valley’s individually.

Click here for additional data file.

Supplemental Information 13 R-code for analyzing impacts of drought on open water in managed wetlands and agriculture using separate models for each year type.

Click here for additional data file.

We thank Catherine Hickey, Kristy Dybala, Kristin Sesser, Nat Seavy, Tom Gardali, and Sam Veloz of Point Blue Conservation Science and Katie Andrews of The Nature Conservancy for feedback and contributions to earlier versions of this work. We are grateful to partners at the Natural Resources Conservation Service for providing Waterbird Habitat Enhancement Program data. This is Point Blue Conservation Science contribution number 2178.

Additional Information and Declarations

Competing Interests

Author Contributions

Data Availability

The authors declare that they have no competing interests. Matthew E. Reiter, Nathan K. Elliott, and Dennis Jongsomjit are employees of Point Blue Conservation Science, and Gregory H. Golet and Mark D. Reynolds are employees of The Nature Conservancy.

Matthew E. Reiter conceived and designed the experiments, performed the experiments, analyzed the data, contributed reagents/materials/analysis tools, prepared figures and/or tables, authored or reviewed drafts of the paper, approved the final draft.

Nathan K. Elliott performed the experiments, analyzed the data, contributed reagents/materials/analysis tools, prepared figures and/or tables, authored or reviewed drafts of the paper, approved the final draft.

Dennis Jongsomjit performed the experiments, analyzed the data, contributed reagents/materials/analysis tools, authored or reviewed drafts of the paper, approved the final draft.

Gregory H. Golet conceived and designed the experiments, contributed reagents/materials/analysis tools, authored or reviewed drafts of the paper, approved the final draft.

Mark D. Reynolds conceived and designed the experiments, authored or reviewed drafts of the paper, approved the final draft.

The following information was supplied regarding data availability:

The raw data and R-code are provided as Supplemental Files.

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
