# Peer review of "Impact of extreme drought and incentive programs on flooded agriculture and wetlands in California’s Central Valley"

_PeerJ, doi:10.7717/peerj.5147_

## Round 0.1 · original submission · Minor Revisions

We have received the reviewers' comments. Your manuscript cannot be published in its present form because some minor corrections are needed. Please, consider all the suggestions provided in the revised version of your manuscript. Additional details have to be included in the materials and methods sections, as well as some unpublished data on supplementary files.

·

Basic reporting

Current article abstract is unstructured, and according to 'standard sections' instructions for authors, requires structuring as follows:

"Headings in structured abstracts should be bold and followed by a period. Each heading should begin a new paragraph.
For example:
Background. The background section text goes here. Next line for new section.
Methods. The methods section text goes here.
Results. The results section text goes here.
Discussion. The discussion section text goes here.
"

A standard heading used by the journal is "Materials & Methods"; the authors use "METHODS", which I presume is appropriate as the research required little in materials.

Experimental design

Re. “Methods described with sufficient detail & information to replicate.”

In general, the methods were well enough described. In a few instances, additional clarity could improve understanding of the methods and study area leading to specific results (regarding which models/model parameter estimates were used to derive certain values, and defining regions and wetland protection status more clearly, defining years analyzed in figure captions). Please see “General comments for the author” and the uploaded annotated pdf document for more details.

Validity of the findings

Re. “Conclusions are well stated, linked to original research question & limited to supporting results.”

In general, the author interpretations are supported by results. In one instance, regarding the effect of rainfall, I did not fully agree with author interpretations.

There is some lack of clarity in text and results relating to table 5 and figures 4-5.

I think the authors could better and more directly focus conclusions on 2 objectives listed on lines 91-94, findings addressing these objectives, and how addressing the objectives advances our understanding for and the future of conservation.

For all of the above more details are provided in “General comments for the author” and the uploaded annotated pdf document for more details.

Additional comments

Dear authors,
Thanks for your contribution!

I think the subject research will add greatly to the current information (or lack thereof) about drought impacts on waterbird habitats and to inform conservation in the Central Valley.

Below are specific comments that I hope will improve the quality of your article.

I inserted additional minor edits directly in the downloadable annotated manuscript text, tables, and figures document to accompany the comments below.

Sincerely,
Elliott Matchett

Abstract starting on line 14- Abstract lacks subheading structure; please review journal’s abstract subheadings-structural requirements to ensure compliance.
Line 27- “non-drought years 2000-2011”- at least 2009 can be identified as a time of drought during this period, so you should probably reword this as something like “a relatively wetter period including years 2000-2011”.
Line 61-62- Because the Cascade Range supplies Shasta Lake (largest surface reservoir draining to the Central Valley) and the upper Sacramento River, I suggest you Identify it here as well. Also, use “ranges” or “mountain ranges” following their names to clarify that the Sierra or Cascades are mountains.
Line 70- reference spelling “Synder” doesn’t match spelling “Snyder” for reference on line 489.
Lines 73-76- Sentence is difficult to read; please reword. Maybe something like “The effects on the distribution of surface water caused by water restrictions, increasing water costs (Howitt et al. 2014), and lack of precipitation, needs to be assessed to understand impacts on waterbird habitat availability.

Lines 83-84- Can you provide info. on depth of flooding for WHEP as you did for BirdReturns? Also, please briefly (in a few words if possible) explain “staged”. I think this means that timing of draining and depth is staggered among fields, but clarification would be nice.

Line 98- Re. “Central Valley Joint Venture planning region (Dybala et al. 2017)”- thank you for providing the citation; however, I suggest that you briefly describe the CVJV, it’s “planning region”, and each’s significance (addressing their importance for conservation and reason for using the subject region as your study area).

Lines 99-101- Typically the San Joaquin Valley is used to refer to the region extending from below the Sacramento-San Joaquin Rivers Delta to the bottom of the Tulare Basin. Therefore, it is very important to rename the region you are calling San Joaquin Valley as “North San Joaquin Valley” or something else. Similarly, change Figure 1 to reflect any renaming change you make. Line 64, text “southern Central Valley (San Joaquin Valley)” also would imply the full southern (below Delta) extent of the Central Valley. Therefore, persons who are familiar with the Central Valley but who do not carefully read lines 99-101, would misconstrue results for your study to pertain to “San Joaquin Valley” and “Tulare Basin” combined. If region names you are using are identical to terminology in Dybala et al. 2017, and thus, you would prefer to maintain current region names, then I still think you should explain in text the distinction between what you are calling San Joaquin Valley from the alternative area extending to the bottom of Tulare Basin that is commonly known as the San Joaquin Valley.

Line 103- Please include “Cascade Range” as well.

Lines 105-107- Although pretty intuitive, you probably need to define water transfers or reword for readers that are unfamiliar with water management, e.g., “often rely on water being transferred for use from the north through contractual agreements (“water transfers”)”.

Line 113- instead of satellites, would it be more accurate to say “satellite imagery” or “imagery from satellites”?

Lines 114-115- Sentence seems awkward as worded with the “5’s” and “8’s”; can you drop the “’s”?: “We used Landsat 5 Thematic Mapper for the period of 2000 – 2011 and Landsat 8 Operational Land Imager and Thermal Infrared Sensor for the period of 2013 – 2015.”

Lines 126-130- please simplify language here and throughout re. “open water” or “water” for crop types, with area/proportion of crop types that were “flooded”, because aren’t your estimates of open water a measure of the flooded area/proportion of each cover type?

Lines 134-138- You present results solely for differences between Sacramento and San Joaquin Valleys (I assume this is only the region above Tulare Basin and doesn’t include Tulare Basin). Are these the only regional differences you investigated?- I assume that they are not based on the literal meaning of region as pertaining to Figure 1. If these 2 regions were the only two compared, then please specify this here and in Figure 5 caption too.

Lines 141-142- good idea weighting by cloud “freeness”; I would guess this would improve the model performance (considering 50% range would include large variation).

Lines 155-156- unclear what is meant by “included estimates of total precipitation within the last two and four weeks”. It becomes much clearer on lines 161-163. Consider deleting “within the last two and four weeks”, or reword to clarify the relationship b/w precip. variables and Landsat data intervals to indicate how they correspond.

Lines 181-184- It is unclear about what is being compared and is differing here: “within year temporal availability of open water might differ in…”. I think within-year availability is differing among year types, but this should be clarified.

Lines 187-188- I think it would be helpful to be more specific actually being compared about “dynamics” in “we compared the dynamics of open water”; and similarly
lines 191-193- “seasonal wetland data” in “We fit separate GAMMs to seasonal wetland data from each region”. You might be alright to leave lines 187-188 as is, while following up with specifics about the data in 191-193.

Lines 213-216- Reword for clarity. Suggested revision: “To account for habitat (>0cm of water) remaining in rice fields upon termination of incentivized flooding, we used data from another study in rice (Point Blue unpublished data) to estimate the average duration that water remained in fields during the period that fields were drained (i.e., drawdown).”

Lines 244-245- “declines in rice (25 and 46%), corn (77 and 81%), and other crops (64 and 71%) (Fig. 2).” Add Table 3 to citation (Table 3, Fig. 2). I see that these values for the 3 crop classes come from Table 3; however, they come from varying models in the table. How did you determine which values to cite here? E.g., for each crop type, are values from the models explaining the most variation, or some other criteria? You should explain your logic in that decision making, and if there isn’t consistent logic, then reevaluate and then briefly explain models and values selected; also, you may wish consider some kind of model-averaging of values to derive single estimates.

Lines 245-247- I don’t completely follow this logic. I agree that average differences between recent drought and other year types is weakened (and is not “statistically significant”, and by the way, should be interpreted with caution) by the positive modeled (additive) effect of seasonal precipitation on open water (10 to 11% more; Table 3, models 2 and 3 vs. model 1). I still think this could be reworded in the context that the positive effect of rainfall is intermittent/transient/periodic to make it more clear about what the rainfall effect means. You don’t model the interaction between year type and precipitation (it’s not shown in either Table 3 or Figure 3) to show support for variable effect of precipitation by year type, so I think the statement “…precipitation…was prominent in non-drought years” must be excluded from text. Here and throughout, I suggest including P > or P< values after using the word significant when regarding statistical significance, so it’s not interpreted otherwise (or alternatively you could state in the methods “significant” refers to statistical significance, so that text is not littered with “P levels of significance.

Lines 251-254, 254-259- see comment for Lines 244-245 relating to selection of model estimates.

Lines 265-268- I don’t understand why estimates of open water in Figure 5 don’t correspond well with the declines cited for Table 5. Figure 5 declines assuming a trend “line” centered within bands would be much closer than the 63-69% (Sac. Valley) and 85% (San Joaq. Valley) difference between recent drought and the other periods. As I read further, also lines 276-278 seem to be in conflict (“In seasonal and semipermanent
wetlands, the nearly 50% reduction in water was largely observed between October
and March (Fig. 4).”). Lines 251-254 (“While seasonally flooded managed wetlands showed significant declines in open water in the recent drought (47 – 59% declines) compared to historic non-drought and average years (Fig. 4; Table 4), changes in semi-permanent wetlands were not significant, though all drought variable coefficients had point estimates that were negative.”), suggest yet another set of values and potential discrepancy with the above and which may need to be put into the appropriate context as well. Perhaps you can reword or clarify to avoid this confusion or prevent persons from making inappropriate comparisons between table 5 and figure 5/figure 4.

Lines 275-276- Re. “The reductions in water in all crops were particularly pronounced October to March and then again in May.” Figure 2 appears to show that except for corn, crops (i.e., rice and “other crops”) tended to comparatively lack open water in recent drought during Nov.-Feb. (corn during Oct-Feb). Sentence in lines 327-328 also seems to be consistent with this idea. Please revise accordingly.

Lines 286-288- I think you can delete sentence “Our model…”. I question whether it adds any additional info. to what follows.

Lines 298-299- awkward structure; reword: “and while between 1 February and 7 March, WHEP provided 31% (min = 15%, max = 48%)”. Perhaps “and WHEP provided 31%(….) during February-7 March.”

Lines 350-352- I think you mean 70% of the available habitat on rice land (not all habitat), right?

Lines 389-402- Please better and more directly focus your conclusions on your 2 objectives listed on lines 91-94, your findings addressing these objectives, and how addressing the objectives advances our understanding for and the future of conservation. I think much of the text (especially lines 390-397) in your conclusions might better fit somewhere within the discussion leaving more space for refocusing your conclusions.

Lines 401-402- Re. “where water management needs to focus in the face of additional, and potentially more extreme, drought events.”, please be more specific about what is meant by “where” because I don’t think that you mean exclusively regional. Thinking about study results you could point to habitats, times of year, and Sac. Valley and/or San Joaquin Valley.

Table 1- in caption, specify the units for accuracy and bias (e.g., percentage of locations accurately classified and percentage difference between ground-verified observations and model predictions of open water). If you are trying to reduce width of the table as implied by the wrapped text for wetland, can you simplify the 3 words for wetland to just “wetland” and accurately define type(s) of wetlands (which I thought was seasonal and semi-permanent) in the caption. Also, you should probably define “other” too as you did in Table 2, so the table can stand alone.

Table 2- in both the caption and in footnote 3, you have “The protection variable was not included in crop type models.” or similar text. You should probably include it in only one of these. I think you could describe the protection variable slightly more so the reader understands what constitutes being protected to allow potentially more flooding of these wetlands. The “**” before this text in the footnote appears to be out of place.

Table 3- You’re missing a value for %, model 2, corn. Although in text line 244, you state “yearly declines”, you should state that models are estimating variable relationships and changes on an annual time scale in tables 3 and 4.

Table 4- Are you able to fit “Protected*Recent Drought” on one line (you only have 2 land cover classes in this table and I see a fair amount of empty space b/w columns).
I think you should add a superscript and footnote for “Covariate” indicating what “Protected*Drought” and “Protected*Recent Drought” represents (i.e., which level of Protected is indicator = 1 that is being expressed).

Figure 2- Figure caption: “Estimates were derived from separate models for each year group…”. You do not indicate which model(s) (of 1-3) or model effects that were used to estimate the relationships presented; please do so. Comments below regarding drought terminology and related text apply to all parts of the manuscript. You classify drought for years 2013-16 as “Extreme Drought” in caption (also in Figs. 4 and 5), but in Tables 2-5 you classify this period as “Recent drought”; be consistent in your terminology of drought periods and I suggest using “recent drought” throughout because “extreme drought” suggests that it was a worse drought than periods of drought during 2000-11 despite providing no such evidence (which I believe exists). If you want to make comparisons based on relative severity of drought because you think that it would help in making conclusions then consider providing supporting information on drought severity to indicate 2013-15 is more severe than drought during 2000-11.

Figure 3- I find the 3 color bands a bit confusing. To resolve this, I have some suggestions. Figure caption: indicated what the bands represent (95% confidence intervals?). Figure legend: if possible, add a third category to the legend that represents the intermediate color where precip. and no precip. bands overlap and indicate as region of overlap between precip. and no precip.

Figures 4 and 5- I think you need to be explicit about the models that these estimates are from; too little info. is provided on the models/model variables. It is unclear which model of 1-6, or if separate models (and what variables) were run just for figure presentation.

In figure 5, if possible please explain the other colors for overlap of the 3 periods, or even perhaps experiment with and different line types for boundaries instead of colors for each year type; different line types may make it easier to see areas of overlap among year types. I think it would be helpful to specify the model variables in the caption. As stated in an earlier comment, if region in the model only compared these two regions and not all 4 regions indicated in Fig. 1 then I think this should somehow be specified; or, if results are actually for Sac. Valley and the combined region of the “San Joaquin Valley” and “Tulare Basin” in Fig. 1(as typically these regions combined is called the “San Joaquin Valley”), then San Joaquin Valley should be defined as such for this figure and in text. This also would imply that you should redefine regions and boundaries in Fig. 1 to reflect this combined southern region.

Figure 6- Clarify which years these data represent.

Review of R script and raw data files:
I was able to open all R script and data (csv) files. R scripts were well documented with extensive comments and identifying the data files that are used in R script code for analysis. Variables in each data file were clearly labeled and variables appeared to be full populated with meaningful values.

·

Basic reporting

This paper is clear, concise, and very well written. Most data are clearly referenced. Figures highlight the data well and are easy to understand. Some minor adjustments can be made to improve both background context for the study as well as transparency.

For readers not familiar with Central Valley water and habitat management, additional information in the introduction regarding habitat management programs and water supply can provide valuable context for later interpretation of results. For example, lines 51-54 could include additional background information regarding the complexity of Central Valley water infrastructure and water allocation, and lines 78-80 could include additional information about the size and scale of incentive programs compared to protected habitats.

There are instances, such as lines 128-130, and lines 214-216, where there is mention of use of unpublished data. While I did not review supplemental material to determine whether those data were provided, I recommend including a reference to supplemental materials or an alternative way to access the data. This will increase the reproduce-ability and transparency of the research.

Experimental design

Most of the methods are well described and appear to be reproduce-able, with some minor exceptions presented here.

There are several instances where additional detail is needed to understand and reproduce analytical methods:

It is explained in lines 114-116 that a change in LandSat sensors occurred between the base period and the drought period, and suggested that ground-truth validation was use to bias correct. However, additional explanation is needed regarding how these sensors differ, and if additional methods were used to avoid step trends in the data.

It is explained in lines 139-141 how images with cloud cover were handled. However, additional information is needed regarding validation procedures to assess these methods. For example, how were random effects and individual observations used?

In lines 207-209, more information is needed to describe how habitat days were computed from bi-weekly open water estimates.

Estimates of open water that overlies various base landcover datasets were used to determine whether areas are considered to be waterbird habitat or suitable for waterbird use. However, in lines 216-218, clarification is needed regarding whether depth and saturation elements of rice habitats were computed as part of this study, used from another study, or whether other assumptions were made to presume presence of these conditions where open water intersects certain landcover types.

Validity of the findings

Most of the conclusions are well stated, and clearly linked either directly to supporting results, or by a combination supporting results and other referenced research. Several instances were noted where discussion did not have a strong backing from statistical results. In other areas, additional discussion of model results would provide benefit, even if negative or inconclusive.

Lines 238-240 describe the criteria for a "reasonable fit" for the GAMM model being indicated by an adjusted R-squared value of 30 to 79 percent. However, in terms of predictive power, adjusted R-squared values of less than 50 seem questionable. This is not a critical issue because a.) these models are mainly used for explanatory power rather than predictive power, and b.) because each model has at least one coefficient with statistical significance (but rarely more than one). However, additional discussion of why low adjusted R-squared values are considered a reasonable fit, because of the study objectives and use of the results (for example), would improve the validity of this claim.

There are several instances in lines 247-263 where interpretive meaning is assigned to coefficients with "marginal significance" (p-value <.1 but >.05). I would urge caution and recommend avoiding any conclusion that these co-variates can explain variation of the dependent variable, or have significant meaning.

Lines 376-379 include a good discussion on the wetland findings in terms of overall declines from drought and between study regions. However, I think there could be additional discussion in this paragraph about what the model results are saying about other driving forces for wetland variability. I saw that several of the models showed that protected status was the significant factor for explaining variation of semi-permanent wetlands, but had a negative coefficient. Is there a meaningful explanation for this result? If there is not, I recommend mentioning this in the discussion.

Additional comments

I greatly appreciate the opportunity to review this paper. This paper presents important findings for improved understanding driving forces behind variability in availability of habitat across the Central Valley, especially given the recent drought. The finding that precipitation drives rice habitat variability in spite of a complex water management system that has time delays from climate events was surprising, and shows that this type of analysis can serve to greatly improve our knowledge in this area. The findings regarding the changes in habitat timing under drought conditions is incredibly informative for inter-annual strategic planning by conservation managers when preparing for future droughts. The findings regarding the contribution of incentive programs are promising and provide evidence for success for potential adaptation strategies in the face of increased drought duration and intensity.

In addition to the review comments above, I have made additional minor comments in the annotated pdf that is attached with this review. Most of these comments are requests to clarify terms and ensure understanding of terminology by broader audiences, as well as some minor suggestions to improve figure and table explanation or content.

---

## Round 0.2 · accepted · Accept

I am pleased to confirm that your paper has been accepted for publication in PeerJ. Thank you for submitting your work to this journal.

#